# Exploring Crowd Travel Demands Based on the Characteristics of Spatiotemporal Interaction between Urban Functional Zones

**Ju Peng** 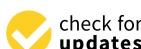, **Huimin Liu**, **Jianbo Tang \***, **Cheng Peng, Xuexi Yang**, **Min Deng and Yiyuan Xu**

Department of Geo-Informatics, Central South University, Changsha 410083, China; daisy_pj@csu.edu.cn (J.P.)
* Correspondence: jianbo.tang@csu.edu.cn

**Abstract:** As a hot research topic in urban geography, spatiotemporal interaction analysis has been used to detect the hotspot mobility patterns of crowds and urban structures based on the origin-destination (OD) flow data, which provide useful information for urban planning and traffic management applications. However, existing methods mainly focus on the detection of explicit spatial interaction patterns (such as spatial flow clusters) in OD flow data, with less attention to the discovery of underlying crowd travel demands. Therefore, this paper proposes a framework to discover the crowd travel demands by associating the dynamic spatiotemporal interaction patterns and the contextual semantic features of the geographical environment. With urban functional zones (UFZs) as the basic units of human mobility in urban spaces, this paper gives a case study in Wuhan, China, to detect and interpret the human mobility patterns based on the characteristics of spatiotemporal interaction between UFZs. Firstly, we build the spatiotemporal interaction matrix based on the OD flows of different UFZs and analyze the characteristics of the interaction matrix. Then, hotspot poles, defined as the local areas where people gather significantly, are extracted using the Gi-statistic-based spatial hotspot detection algorithm. Next, we develop a frequent interaction pattern mining method to detect the frequent interaction patterns of the hotspot poles. Finally, based on the detected frequent interaction patterns, we discover the travel demands of crowds with semantic features of corresponding urban functional zones. The characteristics of crowd travel distance and travel time are further discussed. Experiments with floating car data, road networks, and POIs in Wuhan were conducted, and results show that the underlying travel demands can be better discovered and interpreted by the proposed framework and methods in this paper. This study helps to understand the characteristics of human movement and can provide support for applications such as urban planning and facility optimization.

**Keywords:** spatiotemporal interaction; travel demand; urban functional zones; hotspot detection; frequent pattern mining

## 1. Introduction

With the continuous progress of society and the improvement of people's living conditions, cities have gradually become important places for people's life and production. The growing material and cultural needs of citizens have promoted the differentiation of urban land use functions and evolved into different urban functional zones (UFZs), e.g., residential areas, shopping districts, business areas, entertainment areas, and education areas [1]. As an important part of urban structure, UFZs act as basic geographical units that carry citizens' daily life and socio-economic activities. They bear important geospatial attributes and semantic features of urban land and human activities. Each functional area in urban space is organically connected with each other, rather than an independent individual. They are closely connected with the human movement in the flow of matter, energy, and information, forming the common geographical phenomenon of spatial interaction [2,3]. Spatial interaction reflects important information about citizens' travel patterns, purposes, and urban spatial structures, which are useful for applications such as urban planning,

resource allocation, traffic control, and urban management [4–7]. Currently, many scholars have conducted considerable studies on spatial interaction [8,9], and spatial interaction analysis has become a hot topic in sociology and geographic information science. With the development of location-based services and various mobile sensors, massive trajectory data that record the location, time, movement speed, and status information of human travel in space are collected. The ubiquitous trajectory data contain important information about people's travel characteristics and high-level behavior semantics, providing important data sources and new opportunities for in-depth analysis of people's static or time-dynamic travel patterns and urban spatial structure [10–13].

Currently, there are many studies on spatial interaction patterns between provinces, cities, within provinces and cities [14–16], or urban agglomeration [17], while there is less research on spatiotemporal interaction patterns at the local scale of functional areas [18]. Existing methods mainly focus on the detection of explicit spatial interaction patterns [3], such as clusters in origin-destination (OD) flow data [19–21], without gaining insight into the implicit underlying travel demands of the residents moving between different urban areas [19,22]. Therefore, this paper proposes a framework to explore the residential travel demands based on the characteristics of spatiotemporal interaction between different urban functional zones and gives a case study in Wuhan, China. The main contributions of this study include:

(1) Differently from previous studies, we propose a novel method to discover human travel demands by revealing the spatiotemporal interaction patterns embedded in human mobility activities between different UFZs.

(2) A hotspot interaction pattern mining method is developed to detect the significant frequent spatiotemporal interaction patterns between one urban functional zone and another in the study area. The significance of the spatiotemporal interaction patterns is evaluated using the frequent pattern detection algorithm with support, confidence, and lift metrics. It could provide intuitive insights into the local areas where people significantly gather and the time periods when strong interactions occur.

(3) The characteristics of the hotspot spatiotemporal interaction patterns in terms of travel distance and travel time are analyzed and discussed. Finally, combined with the urban built environment, the driving factors of different spatiotemporal interaction patterns are explained, which are beneficial to reveal the motivation and laws involved in crowd mobility.

The remainder of this paper is organized as follows: Section 2 gives a full review of the related work. We give the details of the framework for spatiotemporal interaction analysis based on UFZs in Section 3. Section 4 describes the study area and data in Wuhan, China, and the experimental results are discussed in Section 5. Finally, we conclude this study and provide future research directions in Section 6.

## 2. Related Work

Spatial interaction analysis is an important topic in human geography. It analyzes spatial interaction from the perspective of man-land coupling and perceives the dynamic evolution of cities [11]. Traditional spatial interaction analysis mainly focuses on static spaces. However, the development of society has promoted the strengthening of connections between different regions. More and more researchers are paying attention to analyzing spatial flows (e.g., crowd flow, information flow, capital flow, and cargo transportation). At present, scholars are devoted to studying urban space from three aspects, including the study of spatial interaction networks [17,23,24], spatial interaction models [9,25,26], and spatial interaction patterns [27,28].

In the study of the spatial interaction model, the gravity model based on the idea of the law of universal gravitation in physics is the most used interaction model. It was introduced into the study of spatial interaction in the 1940s [29]. Early studies mostly used single indicators (e.g., population, GDP) and Euclidean distance to construct a gravity model to study the spatial interaction between regions [30,31]. However, owing to the diversity of

factors affecting spatial interaction, simply establishing a gravity model cannot meet the needs of quantifying spatial interaction. Therefore, various improved gravity models with various indicators or rectified distance measures have been proposed to analyze spatial interaction patterns [32–34]. The gravity model occupies an important position in studying spatial interaction and is widely used. However, due to the limitations of indicators and parameter selection, the gravity model is difficult to adapt to different urban scenarios and application requirements.

Spatial flows within and between cities constitute urban interaction networks, and the analysis of spatial interaction from the perspective of the network has gradually become the focus of many scholars. Rae [35] established a spatial network by graph partitioning and obtained regional interaction relationships by detecting population migration patterns. Referring to the principle of the data field, He et al. [3] constructed a multi-directional interaction network between cities. They analyzed the interaction and dynamic changes of the Wuhan metropolitan areas. Liu et al. [14] and Kempinska et al. [15] built a spatial interaction network based on complex network theory and methods and explored the spatial interactions between regions. These studies mostly concentrate on regional interactions with large spatial scales, such as analyzing spatiotemporal distribution characteristics of crowd flow, information flow, air traffic flow, and cargo flow among inter-provinces or inter-cities.

Recently, with the development of various sensors and the Internet of Things, data related to human activities (e.g., individual movement trajectory, social media check-in data, passenger IC card data) provide opportunities to detect dynamic relationships between regions and to study citizens' travel patterns in urban space. For example, Ouyang et al. [36] analyzed the spatial interaction between urbanization and ecosystem services in China's top ten urban agglomerations from 1995 to 2018 using hotspot analysis and bivariate Moran's Index. Zhu et al. [5] utilized time-series snapshot data of spatial population distributions to detect potential interregional interaction patterns of population migration in China during the Spring Festival. In addition, unlike the traditional point-point, surface-surface interaction analysis methods, Zhang et al. [37] considered the relationship between regions and their neighborhoods and then studied and visualized spatial hotspot flow patterns from the perspective of regional groups. Regarding the travel patterns of citizens, common strategies are to extract key points or origins and destinations (ODs) and then leverage clustering, hotspot detection, spatial scan statistics and machine learning, and other mathematical methods (e.g., matrix factorization) to identify some interesting and meaningful patterns. For example, Park et al. [38] identified hotspots in tourist destinations and spatial interactions across travel places through spatial clustering analysis and sequential pattern mining based on mobile trajectory big data. Zheng et al. [39] extracted ODs from taxi trajectories and discovered interesting areas based on a grid density clustering algorithm. In addition to detecting the travel patterns of citizens at a certain time snapshot, discovering the dynamic changes in citizens' travel patterns over time has also received increasing attention [13,40,41]. Discrete-time modeling and continuous-time modeling are two ways to discover the changes in travel patterns over time [42,43]. Owing to the complexity and observability of travel patterns, discrete-time modeling is the most used in existing research. Jia et al. [44] built multilayer spatial interaction networks with ODs using a discrete method and detected the dynamic spatial patterns of communities. However, a key issue exists regarding how to choose an appropriate and reasonable time interval, which will influence the results and the interpretability of the detected patterns.

Through the review of previous studies, it can be found that current research on spatial interaction has made great progress in theories, methods, and applications. However, there are some issues that still need further research and analysis. On the one hand, existing studies on spatial interaction are mainly focused on the detection of explicit spatial flow clusters or hotspot interaction patterns, with less consideration of the underlying implicit motivation of human mobility, especially at fine scales. On the other hand, research on the detection and interpretation of spatiotemporal interaction patterns from the perspective of

urban functional areas is still insufficient, which can help to reveal the underlying motivation of people traveling from one area to another. Therefore, to address the abovementioned issues, this paper proposes a framework to explore the travel demands of crowds based on the characteristics of spatiotemporal interaction between different urban functional zones.

## 3. Methodology

As we know, the main reasons for citizens to move within cities are usually for work, shopping, medical treatment, school, etc. Discovering the travel demands of human mobility in urban spaces is important for urban planning, traffic management, and service recommendation. Urban functional zones (UFZs) such as commercial, residential, industrial, cultural, educational, medical, and other service areas are the basic units of urban structure. They contain important geospatial attributes and semantic information about land use, which can be used to infer the implicit travel demands of citizens. In this paper, different from existing studies, we use functional areas as the basic analysis units and propose a framework to identify residential travel patterns and travel demands based on the characteristics of spatiotemporal interaction of urban functional areas. The UFZs are identified based on the method in reference [45]. The overview of this study is shown in Figure 1.

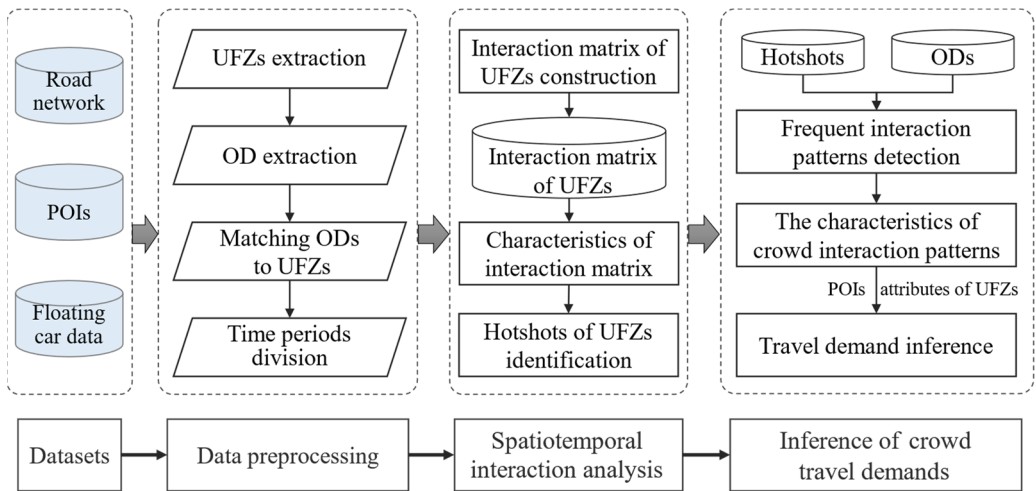

**Figure 1.** Overview of this study (urban functional areas are abbreviated as UFZs).

In the proposed framework, based on the input data, including POI data, floating car data, road networks, and urban functional areas in the study area, we first constructed the spatiotemporal interaction matrix by matching OD flows extracted from the floating car data into the functional areas. Then, we detected the hotspot areas and the frequent interaction patterns to discover the significant pairs of functional areas in different time periods. Finally, based on the discovered spatiotemporal interaction patterns and the semantic features of urban functional areas and POIs, we explored the implicit travel demands of crowds. Details of the framework are described as follows.

### 3.1. Construction of the Spatiotemporal Interaction Matrix Based on Urban Functional Zones

Spatial flows between different functional zones can reflect the spatiotemporal interaction of the city. In this section, we constructed the spatiotemporal interaction matrix of different urban functional zones with ODs extracted from floating car data and analyzed the characteristics of the interaction matrix. The interaction matrix is a square matrix, in which the rows and columns represent one of the corresponding functional zones, and the value of the matrix is the total number of OD flows that point O and point D fall in the UFZ pair, respectively. Then, we explored the characteristics of the interaction matrix with three statistical measures, i.e., spatial interaction strength, travel distance, and travel time of the OD flows in the interaction matrix.

The three measures describe different aspects of OD flows in a statistical sense. Spatial interaction strength refers to the total number of OD flows from one functional zone to another functional zone. The larger the spatial interaction strength, the more the two functional zones are related. By visualizing the spatial interaction strength, we can find the most related functional zones in urban space and the least related functional zones and vice versa. Travel distance and travel time are also two important indicators to describe the characteristics of human mobility and spatiotemporal interaction in urban space. As we know, people would consider distance, time, and cost when traveling. Within some geographical distance and time ranges, many people would like to take a taxi while moving in cities, while beyond a certain range, the number of people taking a taxi will decrease [35]. In the spatiotemporal interaction of UFZs, the interaction strength decreases with the increase in travel distance and travel time, which is also called the distance attenuation effect. To reveal crowd travel patterns at different distance and time range scales, we calculated the statistical distribution of travel distance and travel time of crowds in the study area with the above interaction matrix. Specifically, based on the extracted OD flows, the space distance from one origin point (say i) to its coupled destination point (say j) is calculated as the travel distance $d_{ij}$, and the time interval between these two origin and destination points is counted as the travel time $t_{ij}$. Overall distribution histograms of the travel distances and travel times are used to show the characteristics of crowd travel patterns. From the statistical distributions of travel distance and travel time, we can find the representative travel distance and travel time that people choose the most in the study area. The above information or characteristics are helpful for urban planners, traffic managers, and enterprise sellers to carry out personalized recommendations and services.

### 3.2. Detection of Hotspot Zones in the Spatiotemporal Interaction Matrix

Since different regions have different attractions for crowd flow, it is of great significance to find out which regions have greater attraction to crowd flow to reveal the driving force behind crowd flow. Hence, this paper first uses the constructed spatiotemporal interaction matrix to identify the hotspot zones where people significantly gather, and these zones are further used as core regions to extract the hotspot interaction patterns. In this paper, hotspot zones are areas with strong interaction and frequent crowd activities that can reveal citizens' travel demands and purposes from a local perspective. The inflows and outflows of one region embody the interaction intensity of citizens, and they reflect the closeness of the connection between the region and other regions. Regions with a significantly greater number of inflows and outflows than other regions are regarded as hotspots. In this paper, as shown in Figure 2, when a hotspot is surrounded by other hotspots, the hotspot is defined as a hotspot pole (shown in red), such as regions A and B in Figure 2a.

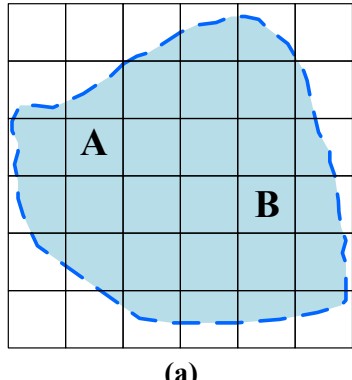
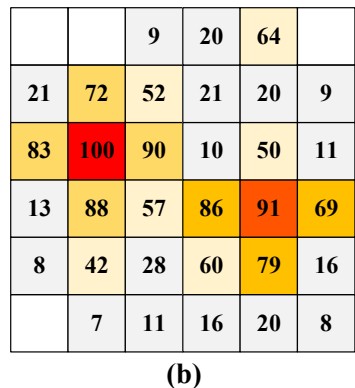

(a)　　　　　　　　　(b)

**Figure 2.** Illustration of the hotspot poles. (**a**) Regions divided by a grid; (**b**) The value of each region represents the total number of crowds moving into and out of the region.

In the literature, Getis-Ord $G_i^*$ [46,47] is a commonly used method to detect significant cold spots and hot spots in spaces. The $G_i^*$ statistic measures if there is statistical evidence of clustering of space units with high attribute values, and the $G_i^*$ statistic is calculated as:

$$G_i^* = \frac{\sum\limits_{j=1}^{n} w_{i,j} x_j - \overline{X} \sum\limits_{j=1}^{n} w_{i,j}}{S \sqrt{\frac{[n \sum\limits_{j=1}^{n} w_{i,j}^2 - (\sum\limits_{j=1}^{n} w_{i,j})^2]}{n-1}}}$$

$$\overline{X} = \frac{\sum\limits_{j=1}^{n} x_j}{n}, \; S = \sqrt{\frac{\sum\limits_{j=1}^{n} x_j^2}{n} - (\overline{X})^2} \tag{1}$$

where $x_j$ is the total number of inflows and outflows of functional zone j, n is the total number of functional zones in the study area, $w_{i,j}$ is the spatial weight between functional zone i and functional zone j, which is defined as:

$$w_{i,j} = \begin{cases} 1, & \text{if functional i and j are first} - \text{order adjacent} \\ 0, & \text{otherwise} \end{cases} \tag{2}$$

In this paper, we take the total number of inflows and outflows of a region (i.e., the functional zone) as its attribute value. We then use the Getis-Ord $G_i^*$ statistic to detect interactive hotspot poles in different time periods, such as morning peak hours and evening peak hours. The hotspot poles are highly active functional zones with strong connections with other zones, which will be used in the next step to discover spatiotemporal interaction patterns between different functional zones.

### 3.3. Detection of Frequent Spatiotemporal Interaction Patterns and Travel Demands Inference

The hotspot poles reflect the active functional zones with many crowds flowing in and out, which are usually important and meaningful areas to which we should pay more attention. However, hotspot pole detection cannot directly discover the functional zone pairs with strong spatiotemporal interactions from a flow perspective. Therefore, we transformed the problem of detecting the functional zone pairs with strong or frequent interactions into a task of frequent spatial association pattern mining. We extracted the frequent interaction patterns using an improved spatial association mining approach. Spatial association mining is a commonly used method to discover the associations between item sets in a spatial database [48]. In our study, we focused on the OD flows where O points or D points fall in the detected hotspot poles. These OD flows are called candidate OD flows in this paper. To obtain interaction pairs of functional zones that have frequent connections at a certain period, we first constructed the interaction matrix of different functional zones using the focused OD flows, and the construction of the interaction matrix is introduced in Section 3.1.

Here, we present an example of such an interaction matrix, as shown in Figure 3a. The types of functional zones include Administration and public service areas (A), Residential areas (R), Industrial areas (I), Green spaces and square area*s* (G), and Transportation areas (T). The values in the interaction matrix are the number of OD flows where origin points belong to the same type of functional zone and destination points belong to another same type of functional zone. In conjunction with this interaction matrix, we give the basic core concepts involved in frequent pattern mining. They are defined as follows:

Firstly, let $D$ be a set with $N$ candidate OD flows, $OD_{AB}$ be the OD flows that originate from the functional zone type A and end at the functional zone type B, $OD_A$ be the OD flows that originate from the functional zone type A and $OD_B$ be the OD flows that originate from the functional zone type B. Then define $T$ as a set of candidate OD flows and $T \subset D$.

**Definition 1.** itemsets: Itemsets are formed by function type pairs associated with candidate OD flows.

**Definition 2.** support: Support is defined as the probability of OD flows that belong to the same functional zone type at the origin points and another same functional zone type at the destination points, which is calculated as Equation (3). Minimum support is a threshold used to filter out itemsets that are not significant enough in the statistical sense [47].

$$\text{support}(A \rightarrow B) = P(AB) = \frac{|\{T : OD_{AB} \subseteq T, T \in D\}|}{|D|} \qquad (3)$$

Different functional zone pairs can also be obtained directly by normalizing the interaction matrix using the total number of OD flows. As shown in Figure 3b, it is called the support matrix in our paper.

**Definition 3.** confidence: It can be defined as the conditional probability of item A and item B. In our study, it is defined as the ratio of the number of $OD_{AB}$ to the number of $OD_A$. It is calculated as Equation (4).

$$\text{confidence}(A \rightarrow B) = P(B|A) = \frac{|\{T : OD_{AB} \subseteq T, T \in D\}|}{|\{T : OD_A \subseteq T, T \in D\}|} \qquad (4)$$

We can also normalize the values in the support matrix by row (or column, depending on which axis, horizontal or vertical, is considered as the origin functional zone type) to get the confidence matrix, which is presented in Figure 3c.

**Definition 4.** lift: It is defined as the ratio of confidence(A → B) to the probability of item B (calculated as Equation (5)). The lift can reflect the correlation between item A and item B. It is a valid strong association only if the lift of the frequent pattern exceeds 1. The lift matrix can be obtained by normalizing the confidence matrix according to the column values of the support matrix (as opposed to the confidence matrix), as Figure 3d shows.

$$\text{lift}(A \rightarrow B) == \frac{P(B|A)}{P(B)} = \frac{|\{T : OD_{AB} \subseteq T, T \in D\}|/|\{T : OD_A \subseteq T, T \in D\}|}{|\{T : OD_B \subseteq T, T \in D\}|/|D|} \qquad (5)$$

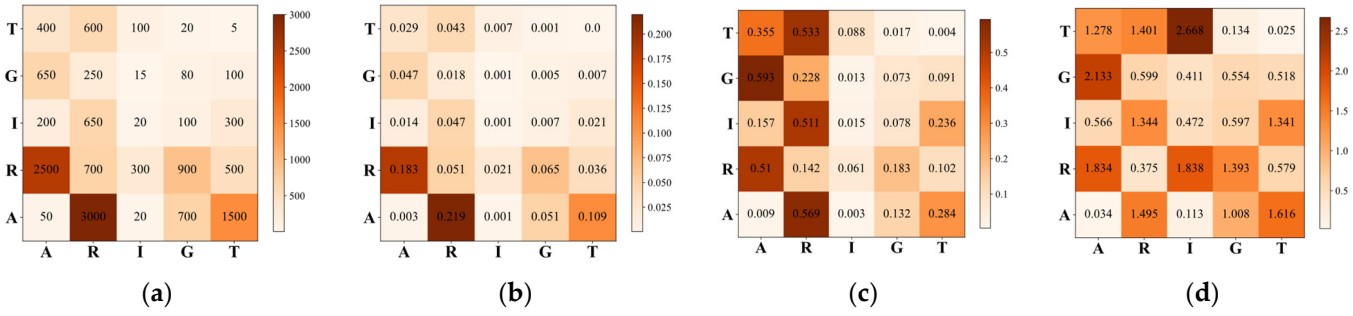

**Figure 3.** Illustration of the (**a**) OD matrix, (**b**) support matrix, (**c**) confidence matrix and (**d**) lift matrix.

Based on the support matrix, confidence matrix, and lift matrix calculated by Equations (3)–(5), we applied the Apriori algorithm to detect the frequent interaction patterns of the functional zones in different time periods with support, confidence, and lift as measures. With the detected frequent interaction patterns, we further try to explain the motivation and travel demands of the interaction patterns by associating the background geographical environment and the semantic features (i.e., land use) of the zones where the origin and destination pairs are within. For example, suppose one hotspot spatial interaction shows a strong connection between a residential area (say Ri) and the commercial and shopping area (say Cj). In that case, it is more likely that most people travel from Ri to Cj with the intention of going shopping. Since urban functional zones act as the basic units of people's socio-economic activities, the functional attributes associated with the origin and destination zones may indicate the implicit motivation and travel demands of human

mobility. Based on this thought, by associating the exterior spatiotemporal interaction patterns with the functional zones, we can effectively interpret the main demands of people moving from one area to another.

## 4. Study Area and Data

Wuhan, known as the thoroughfare of nine provinces and the economic source of the central region, has developed a commercial economy, advantageous geographical location, prosperous science and education culture, and abundant data, which have promoted the development of urban research. As shown in Figure 4a, regions within the Third Ring Road, Wuhan, China, are selected as the study areas in this paper. We first partitioned the study area into different functional zones with a hierarchical fusion method that considers landscape features and human activities [45]. The datasets used to identify functional zones include the road network from OpenStreetMap, POIs (Points of Interest), and trajectory data. To ensure the reliability of the road network data, we performed topology checks and modifications (e.g., rectifying connectivity issues and eliminating pseudo nodes and suspended nodes), road consistency checks, and compared the road network with authoritative maps for reliability verification. The identified functional zones are shown in Figure 4b. For the sake of brevity, in this paper, Residential areas are abbreviated to R, Administration and public service areas are abbreviated to A, Commercial and business areas are abbreviated to C, Industrial areas are abbreviated to I, Transportation areas are abbreviated to T, and Green spaces and square areas are abbreviated to G in Figure 4b and in the following content. The types of functional zones used in our study include both single functional zones (e.g., A, C, and I) and mixed functional zones such as the Administration and public service areas and Residential areas mixed functional zones (denoted by {A, R}).

After obtaining the functional zones in the study area, we used the trajectory data for the analysis of spatiotemporal interactions. We collected a total of 1.65 million trajectories from 8141 taxis from 4–10 May 2014. The main fields of the trajectories include taxi ID, time, longitude, latitude, direction, speed, and passenger-carrying status. We deleted the noise trajectories, as well as trajectories with abnormal velocities and positions. After that, we extracted the pick-up point (origin) and drop-off point (destination) pairs, i.e., OD flows according to the vehicle's passenger-carrying status and speed threshold. The OD flows visualized in the form of a network are shown in Figure 5. By considering the difference in crowd travel patterns on weekdays and weekends and the similarity of the crowd travel patterns on the same type of days, as shown in Figure 6, we selected OD flows on a weekday (Tuesday, 6 May 2014) and a weekend (Saturday, 10 May 2014) for our experiments, respectively.

Since people traveling on a day also has significant temporal dynamics, we divided one day into five time periods by Q-type clustering [49]. The weekday is divided into five representative time periods, including an early morning period (0:00–6:59), a morning peak period (7:00–8:59), a daytime working period between the morning and evening peaks (9:00–15:59), an evening peak period (16:00–18:59), and an evening period (19:00–23:59). The weekend is divided into five representative time periods, including an early morning period (0:00–6:59), a morning peak period (7:00–9:59), a daytime period (10:00–15:59), an evening peak period (16:00–17:59), and a nighttime period (18:00–23:59). OD flows are aggregated according to the divided time periods of weekdays and weekends, and then OD flows in different time periods are used to detect the crowd spatiotemporal interaction patterns in these time periods, respectively.

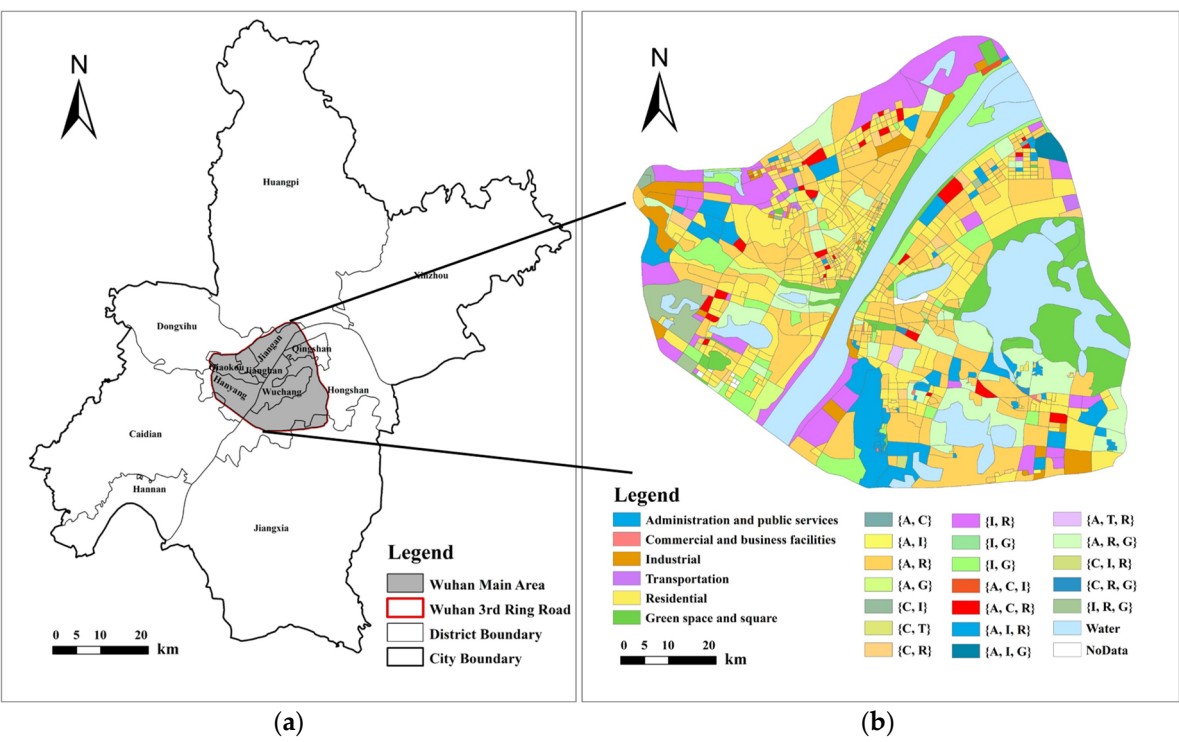

**Figure 4.** Study area and the visualization of urban functional zones of (**a**) study area and (**b**) urban functional zones.

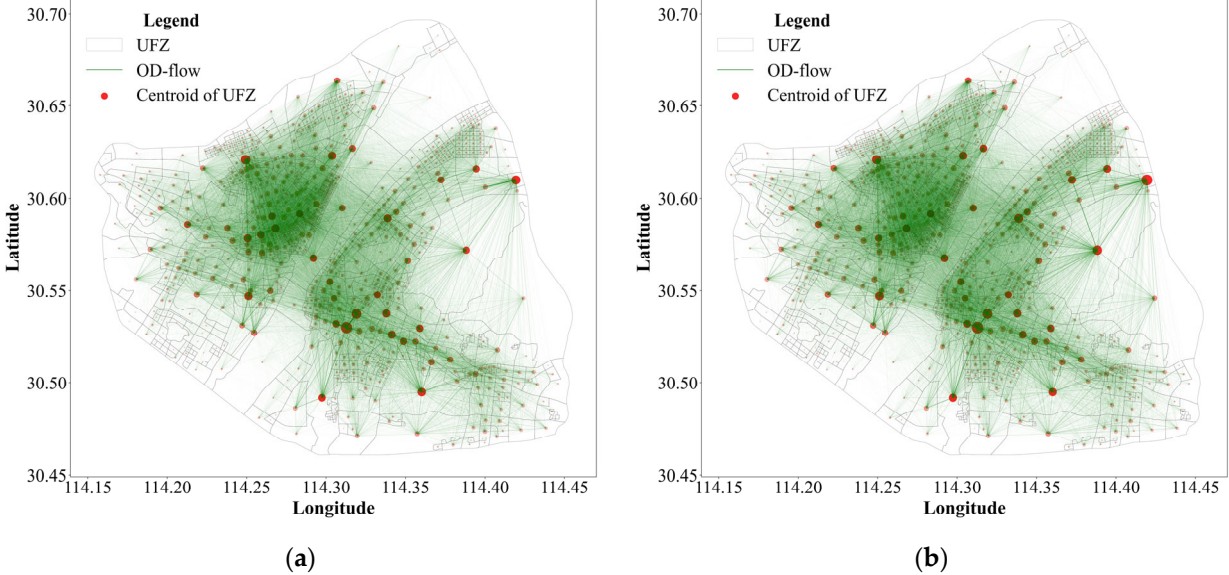

**Figure 5.** OD flows of a (**a**) weekday and (**b**) weekend. The nodes in red are the centroids of functional zones, and the size of the nodes represents the total flow in and out of the functional zones. The line widths of the edges in green denote the amount of the flow from the origin point to the destination point.

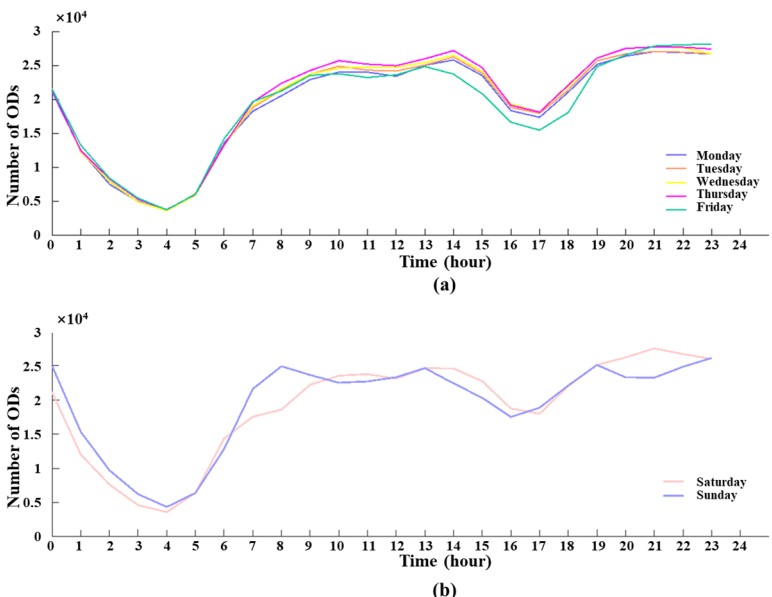

**Figure 6.** The temporal distributions of all-day OD flows on (**a**) weekdays and (**b**) weekends in the study area.

## 5. Results and Discussion

### 5.1. Spatiotemporal Interaction Matrix of Urban Functional Zones

According to the workflow described in Section 3.1, the interaction matrixes on weekdays and weekends are shown in Figure 7. The interaction matrix reflects the strength of the interactions between different functional zones. It can be found that the interaction patterns of different functional zones on weekdays and weekends are similar in Wuhan, China. Regarding the same type of functional zone pairs, the weekend has more interactions than the weekday, indicating that citizens are more active on weekends. Then, excluding interactions of the same type, the top three interaction types are {R → A, R}, {A, R → R}, and {A, R, G → A, R}, both on weekdays and weekends, which is consistent with the fact that 'Residential areas' and 'Administration and public service areas' are the most distributed in the city. Compared with weekdays, the total interaction amount of 'Residential areas' and 'Residential areas containing Industrial type' is relatively small on weekends. In contrast, the interaction amounts of 'Commercial and business areas' and 'Green space and square areas' have increased, which conforms to common sense that people prefer resting, shopping, and taking a trip during the weekend.

Then, we analyzed the characteristics of the travel distance and travel time of citizens in urban space. As shown in Figure 8, the overall distribution of citizens' travel distances on weekdays and weekends is also similar in the study area. The proportion of travel distance within 5 km on weekdays is 75.7%, which is about 3.9% higher than that on weekends, and the proportion of travel distance within 5–10 km is about 19.4% and is approximately 3.5 higher than that on weekends. As for travel distances over 10 km, the proportion on weekdays is about 0.5% lower than that on weekends. Apparently, both weekdays and weekends are dominated by short trips within 5 km, while the weekend has a higher proportion of medium and long trips beyond 5 km than weekdays.

In terms of travel time, the proportion of travel time within 10 min is about 64% on a weekday, and the weekend is about 43% which is significantly lower than a weekday. The proportion of travel time within 10–30 min on a weekend is about 26%, while 44% on a weekend. The proportion of travel time on a weekend with more than 30 min is about 3% higher than that on a weekday. It could be seen that the travel time on a weekday is concentrated within 10 min, and the travel time on a weekend is concentrated within 30 min, and a travel time of more than 30 min accounts for a small proportion on both weekdays and weekends. The above patterns of travel distance and travel time are related

to the phenomenon that the commuting pattern is the main pattern on weekdays, and the distance between work and residence is usually relatively short.

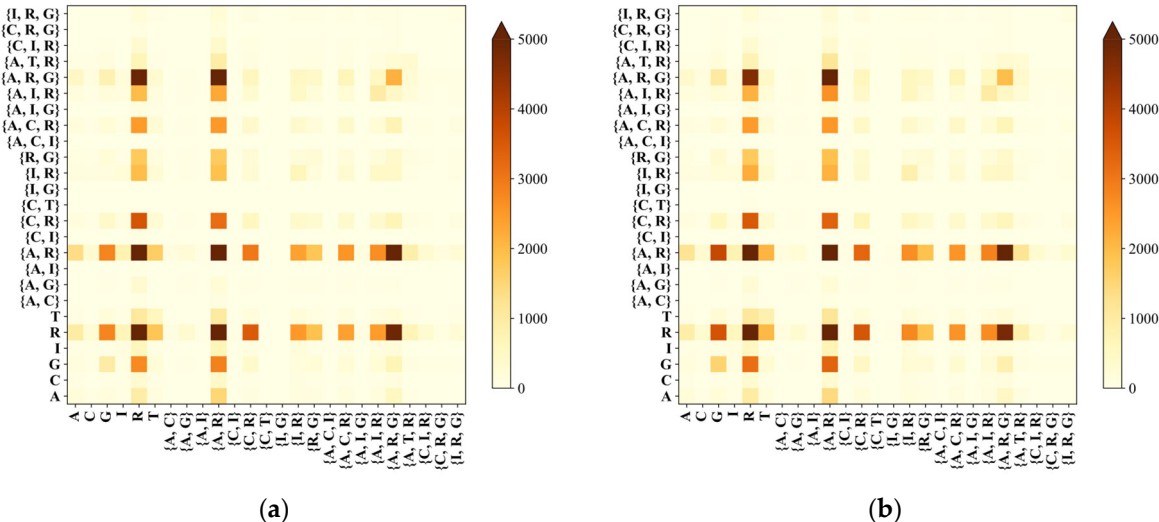

(**a**)                                                                                          (**b**)

**Figure 7.** Interaction matrix of functional zones on a (**a**) weekday and (**b**) weekend.

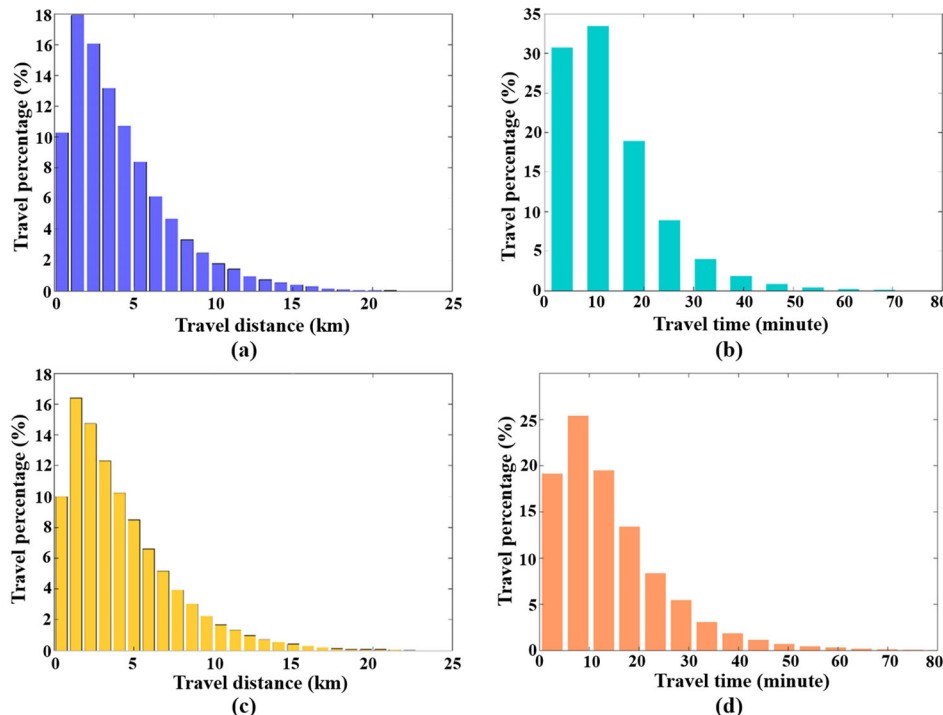

**Figure 8.** Overall distribution histograms of the travel distance and travel time in the study area: (**a**) travel distance distribution on a weekday, (**b**) travel time distribution on a weekday, (**c**) travel distance distribution on a weekend, (**d**) travel time distribution on a weekend.

## 5.2. Results of the Hotspot Poles

The hotspot poles in different time periods obtained using the $G_i^*$ statistic are shown in Figures 9 and 10. Here, we regarded the functional zones with a confidence greater than 90% (i.e., $G_i^* > 1.65$) as the hotspot poles. According to the hotspot detection results, the spatial distribution of hotspot poles in each divided period on a weekday and weekend is similar, mainly distributed in the inner circle of the main urban area, showing a ring shape. Combined with the functional zone type attribute of the hotspots, it can be found that the types of hotspots are mainly mixed functional zones, including {A, R}, {A, R, G},

{A, C, R}, {R, G}, {T, R}, and so on. This reveals that the trend in urban development is towards mixed functional use and development, which can provide diversified services and meet multiple requirements of citizens (e.g., working and entertainment, residence and working, residence and green spaces) through only one functional zone. These mixed functional zones providing diversified services are mainly distributed or near the urban business districts. The hotspots of business districts identified in our study include Xudong, Wuguang, Jiedaokou, Zhongjiacun, and other less prominent business districts. They maintain a high-heat pattern throughout the day. These areas usually have multiple functions and can provide business and public service functions during the day and turn into business and entertainment places at night. Overall, in the early morning period, morning peak period, and daytime period, the distribution of hotspot poles on a weekday is wider. In contrast, in the evening peak period, the distribution scope of hotspot poles on a weekend exceeds that on a weekday, indicating that the urban vitality at night on a weekend is higher than that on a weekday and is relatively lower during the daytime.

In addition, the daily interaction amount of several major transportation hubs in the urban area (i.e., Wuhan Station, Wuchang Station, and Hankou Station) in the early morning period is small, and no hot spots are reflected. The interactions begin to increase after the morning peak, reaching the peak in the daytime and evening peak and falling back to a similar level to the early morning period at night, which is consistent with citizens' daily travel during the day. In addition, owing to the low number of trains running at night, the flow of passengers to and from the station is significantly lower than in the daytime. Overall, the major transportation hubs during the main hours of the day are extremely active and frequently interact with other regions, which can also reflect that another trend of urban development is transit-oriented. The more developed and improved the urban transportation, the greater the vitality of the city. This is because well-developed urban transportation can provide easy access for daily travel or tourist trips, thus increasing spatial interaction within the city.

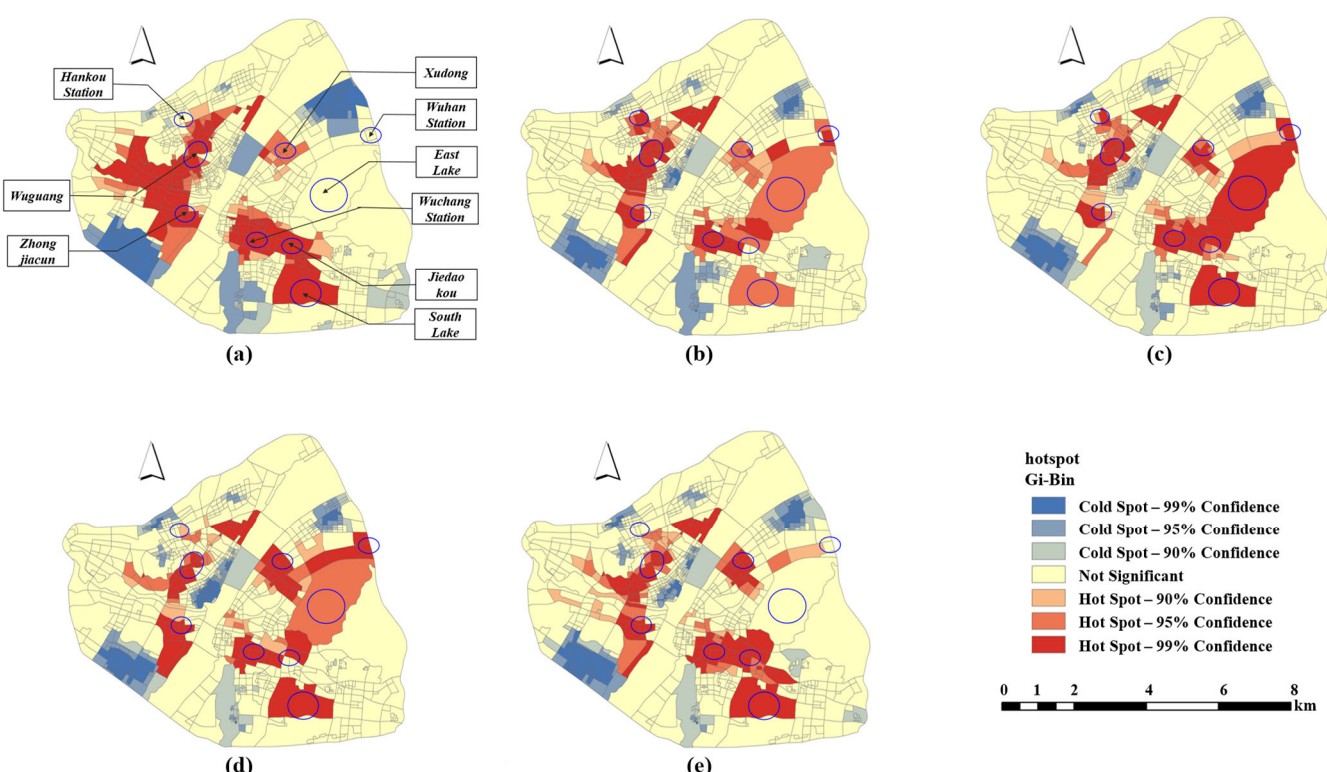

**Figure 9.** Distributions of hotspots in different time periods on a weekday: (**a**) the early morning period (0:00–6:59), (**b**) the morning peak period (7:00–8:59), (**c**) the daytime working period (9:00–15:59), (**d**) the evening peak period (16:00–18:59), (**e**) the nighttime period (19:00–23:59).

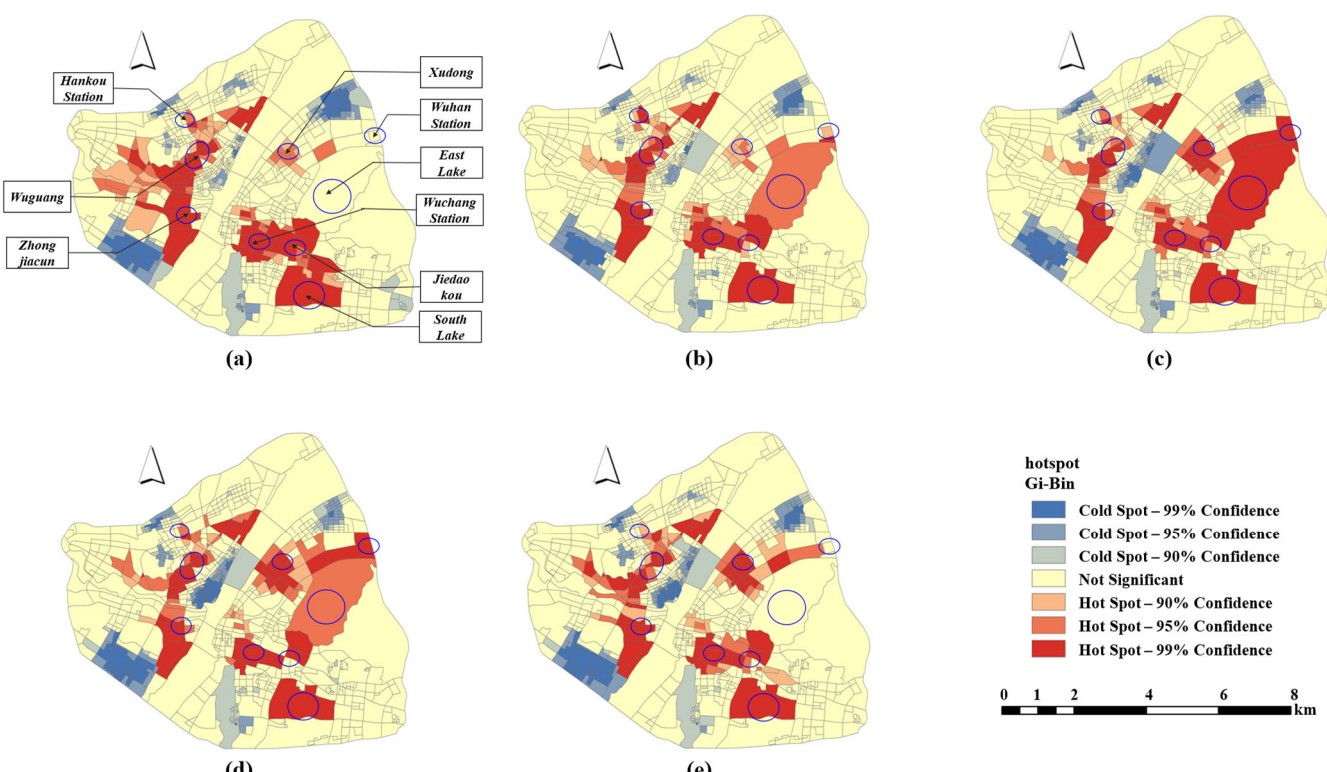

**Figure 10.** Distributions of hotspots in different time periods on a weekend: (**a**) the early morning period (0:00–6:59), (**b**) the morning peak period (7:00–9:59), (**c**) the daytime period (10:00–15:59), (**d**) the evening peak period (16:00–17:59), (**e**) the nighttime period (18:00–23:59).

### 5.3. Results of Spatiotemporal Frequent Interaction Patterns

To detect frequent interaction patterns, we set the minimum support to 0.009 according to our experience and experiments and chose the frequent interaction patterns with a lift over 1. We finally discovered a total of 22 frequent interaction patterns of hotspots on a weekday and 24 frequent interaction patterns on a weekend. First, the frequent interaction results in different time periods are visualized with chord diagrams which can describe the connection between different nodes (i.e., different function types). Different colors in chord diagrams represent different function types. The arc length in the outer circle represents the number of times the function type occurs in the interaction. The thickness of the chord connected between the arcs represents the interaction amount between the functional types. Then, we reported the detected frequent interaction patterns in Table 1, with two typical time periods on a weekday as examples. We also mapped these patterns to the map to analyze the spatial distributions of different frequent patterns. The detailed analysis is below.

As shown in Figures 11 and 12, 'Residential areas' (R), 'Administration-Residential mixed areas' ({A, R}), and 'Administration-Residential-Green space mixed areas' ({A, R, G}) account for the largest proportions on both weekdays and weekends. These three types of functional zones interact most frequently with similar functional zones or other functional zones. 'Green space and square areas' (G) and Green space and square areas mixed with other types of functional zones also occur at all time periods on both days. The proportion of 'Green space and square areas' (G) interacting with other functional zones is greater during the first three time periods on weekends than that on weekdays, while during the last two time periods on weekends, this proportion is smaller than that on weekdays. As for the total types of frequent interaction patterns, the weekend has more diverse types of the frequent interaction patterns than the weekday. It is related to the fact that citizens will carry out more colorful activities on weekends with more various travel intentions. On weekdays, the overall functional interaction pattern reflects the tidal phenomenon.

The flow from other functional zones into 'Residential areas' (R) and 'Administration-Residential mixed areas' ({A, R}) is larger in the morning period and evening peak period than in other periods. This phenomenon is related to citizens' daily commuting patterns. In these two periods, most crowds return to their residence from the workplace, business, or entertainment places. From 7 o'clock, citizens start to work from 'Residential' (R) and 'Administration-Residential mixed areas' ({A, R}) to 'Transportation' (T), 'Commercial' (C), and 'Industrial' (I) areas. Especially, 'Transportation' (T), which only appears in the morning peak hours and the daytime working period on weekdays, has frequent interactions with other types of functional zones during the first three time periods (i.e., early morning period, morning peak period, and daytime period) on weekends. Furthermore, most of the crowds returning to the residence during the early morning period are from Commercial zones, Administration and public service zones, and industrial functional zones, which may be caused by overtime. The proportion of 'Residential areas' (R) on weekdays is significantly higher than on weekends.

**Table 1.** The frequent interaction patterns in the morning peak hours and evening peak hours on a weekday.

| Time Period | Pattern ID | Interaction Pattern | Support | Confidence | Lift |
|---|---|---|---|---|---|
| The morning peak hours (7:00–8:59) | 1 | R → R | 0.1068 | 0.3128 | 1.0639 |
| | 2 | {A, R} → {A, R} | 0.1024 | 0.3064 | 1.0112 |
| | 3 | *** {A, R} → R | 0.0985 | 0.2951 | 1.0037 |
| | 4 | *** R → {A, R, G} | 0.0253 | 0.0741 | 1.0036 |
| | 5 | {A, R, G} → {A, R} | 0.0216 | 0.3099 | 1.0227 |
| | 6 | R → G | 0.0165 | 0.0483 | 1.0518 |
| | 7 | *** R → T | 0.0157 | 0.0460 | 1.0090 |
| | 8 | *** {A, R} → T | 0.0156 | 0.0470 | 1.0251 |
| | 9 | R → {C, R} | 0.0145 | 0.0424 | 1.0402 |
| | 10 | {A, R} → {C, R} | 0.0139 | 0.0415 | 1.0198 |
| | 11 | {A, R} → {A, C, R} | 0.0118 | 0.0353 | 1.0386 |
| | 12 | G → {A, R} | 0.0104 | 0.3197 | 1.0549 |
| | 13 | {A, R} → {A, I, R} | 0.0103 | 0.0307 | 1.0226 |
| The evening peak hours (16:00–17:59) | 1 | {A, R} → {A, R} | 0.0992 | 0.3160 | 1.0019 |
| | 2 | R → R | 0.0986 | 0.3190 | 1.0425 |
| | 3 | *** {A, R} → {A, R, G} | 0.0237 | 0.0756 | 1.0732 |
| | 4 | {A, R, G} → {A, R} | 0.0231 | 0.3281 | 1.0401 |
| | 5 | R → {C, R} | 0.0170 | 0.0548 | 1.1024 |
| | 6 | {A, R} → {C, R} | 0.0161 | 0.0513 | 1.0319 |
| | 7 | *** {C, R} → R | 0.0157 | 0.3287 | 1.0741 |
| | 8 | *** G → R | 0.0138 | 0.3388 | 1.1071 |
| | 9 | G → {A, R} | 0.0132 | 0.3238 | 1.0265 |
| | 10 | R → G | 0.0127 | 0.0410 | 1.0976 |
| | 11 | {A, R} → {A, I, R} | 0.0127 | 0.0403 | 1.0070 |
| | 12 | {A, R} → {A, C, R} | 0.0107 | 0.0342 | 1.0092 |
| | 13 | *** {A, C, R} → R | 0.0101 | 0.3138 | 1.0254 |

(*** indicates patterns that do not appear in another time period.).

As reported in Table 1, the frequent interaction patterns are not exactly symmetrical in the morning peak hours and evening peak hours. The patterns of R → R, {A, R} → {A, R}, R → G, and {A, R, G} → {A, R} appear in the morning peak hours and have symmetrical patterns in the evening peak hours, which reflects the commuting behavior of many citizens. However, the patterns of {A, R} → R, R → {A, R, G}, R → T, and {A, R} → T only appear in the morning peak hours, without symmetrical patterns in the evening peak hours. The reason for this may be that the total interaction amount during the morning peak hours is much higher than that during the evening peak hours. Citizens who work overtime will return to their residences later, and citizens who leave work early may also go to some leisure places, such as parks and squares. Then we analyzed the spatial distributions of

the frequent interaction patterns in the morning peak hours on a weekday. According to the function type that the patterns contained, we divided the frequent interaction patterns into five categories. The spatial distributions of these five categories are shown in Figure 13. In Figure 13, the lines with different colors denote different types of frequent interaction patterns. The red and blue points are the centroids of the urban functional zones, where the red indicates that the inflow is greater than the outflow and the blue indicates that the outflow is greater than the inflow. The size of the points represents the total amount flowing into and out of the functional zones.

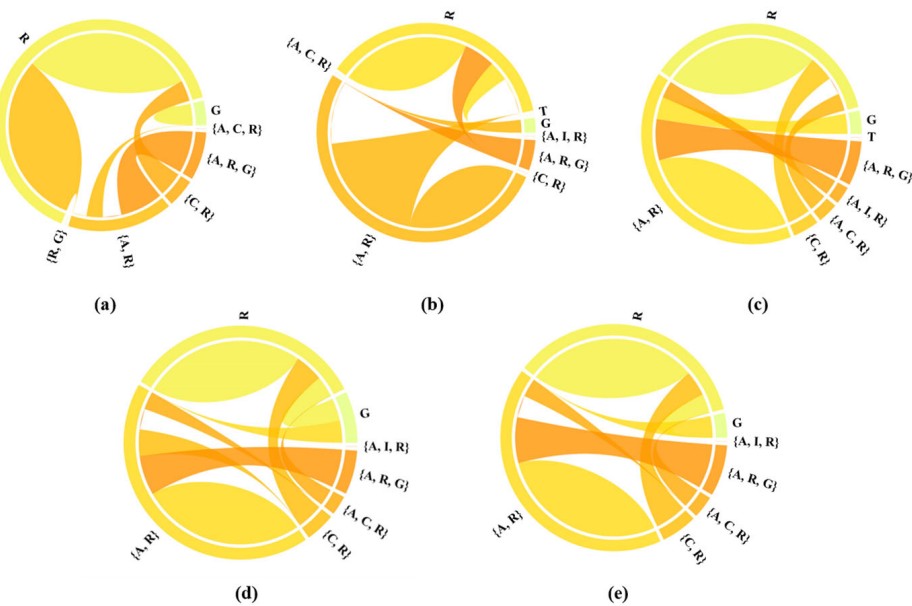

**Figure 11.** Chord diagrams of the frequent interaction patterns in different time periods on a weekday of (**a**) 0:00–6:59, (**b**) 7:00–8:59, (**c**) 9:00–15:59, (**d**) 16:00–18:59, and (**e**) 19:00–23:59.

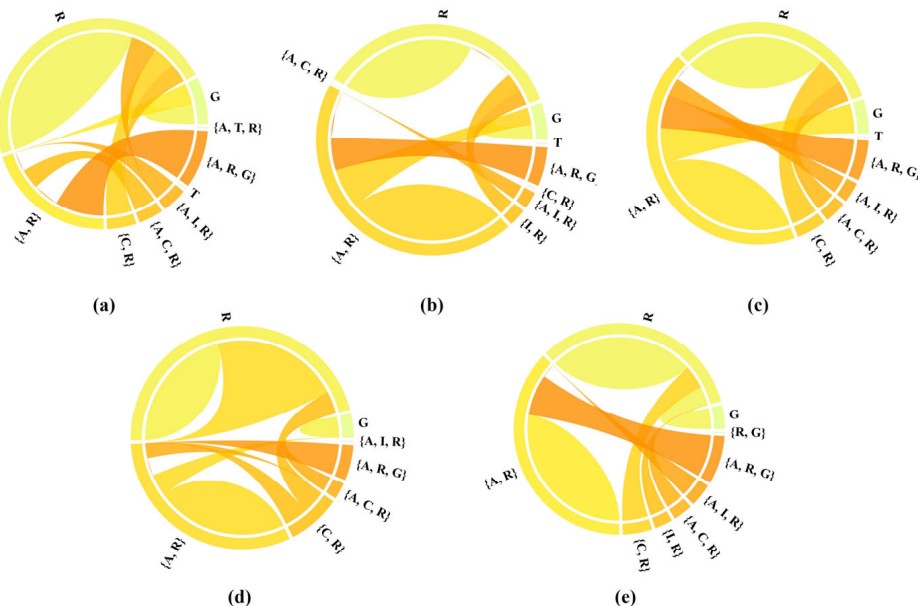

**Figure 12.** Chord diagrams of the frequent interaction patterns in different time periods on a weekend of (**a**) 0:00–6:59, (**b**) 7:00–9:59, (**c**) 10:00–15:59, (**d**) 16:00–17:59, and (**e**) 18:00–23:59.

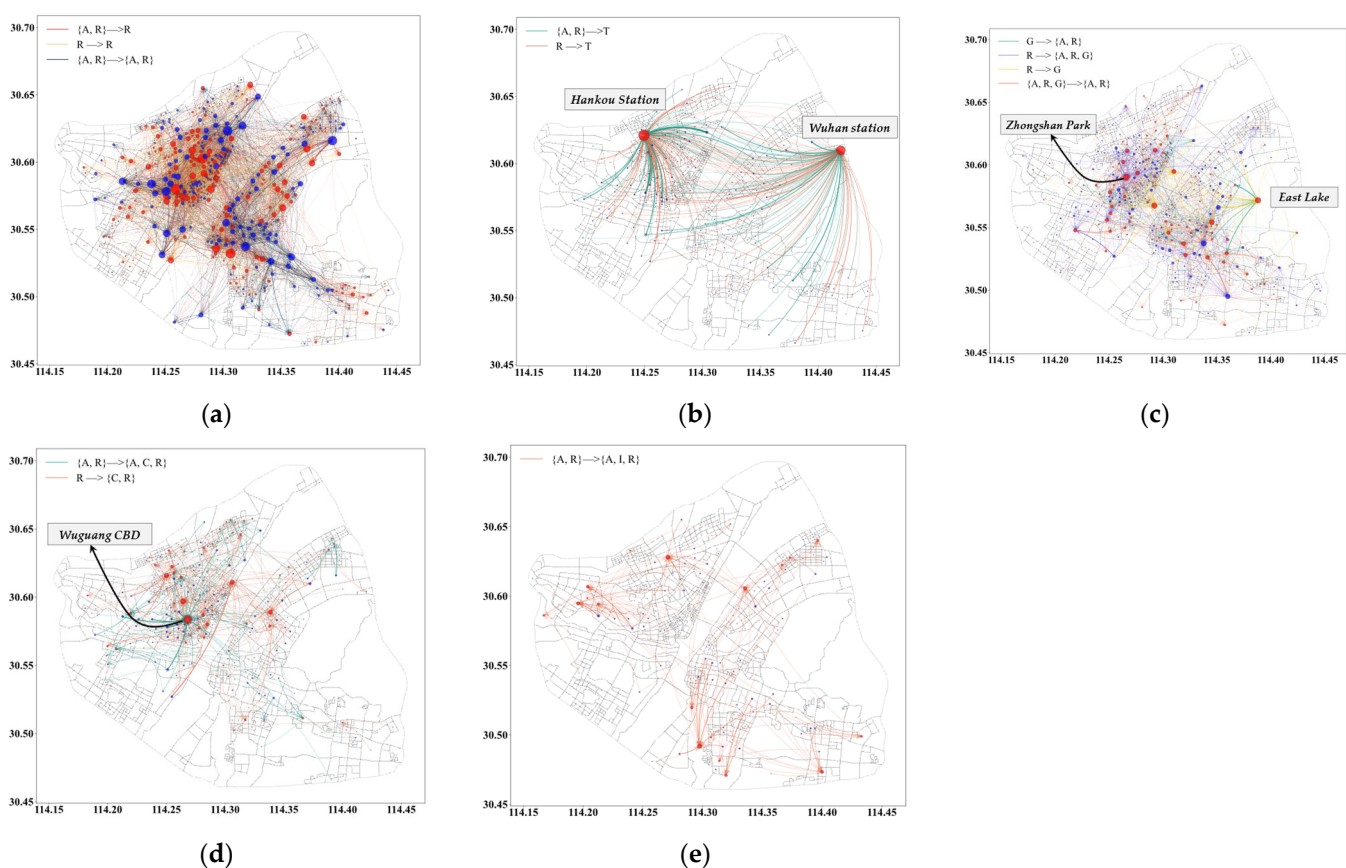

**Figure 13.** The spatial distributions of different frequent interaction patterns in the morning peak hours on a weekday. The nodes in red represent functional zones with inflow exceeding outflow and the nodes in blue represent functional zones with outflow exceeding inflow. (**a**) Category 1 (**b**) Category 2 (**c**) Category 3 (**d**) Category 4 (**e**) Category 5.

It can be seen from Figure 13 that the spatial distributions and the interaction amount of these patterns are different. As for category 1, patterns that contain 'Administration and public service areas' (A), there are quite a few functional zones with larger inflows and outflows. They can well reflect citizens' daily travel patterns, leaving their residences for work. Category 2 consists of patterns containing 'Transportation areas' (T). As shown in Figure 13b, Hankou Station and Wuhan Station are two important transportation hubs for citizens to make cross-city or cross-province trips, thus attracting a large amount of traffic flow from all over Wuhan. Regarding category 3, patterns that contain 'Green space and squares' (G), the main gathering places of the crowds are the various attractions in the city, such as East Lake and Zhongshan Park. Most of the trips in the type of patterns are short-distance travel from the length of OD flows. The fourth category comprises patterns containing 'Commercial and business areas' (C). Most of the OD flows are located in the core urban area east of the Yangtze River. As for the last category, patterns containing 'Industrial areas' (I), the OD flows mainly flow into the fringes of the study area, where the emerging industrial zones are likely to be located.

## 5.4. Characteristics of Travel Patterns and the Inference of Travel Demands

Based on the results of frequent interaction patterns of urban functional zones, we infer and analyze the underlying intentions involved in citizens' travel patterns by exploring the characteristics of travel distance and travel time of citizens in Wuhan city. Figure 14 illustrates the travel distance characteristics of the detected frequent interaction patterns on weekdays and weekends. First, the number of frequent interaction patterns on the weekend is greater than that on weekdays, which reflects the general phenomenon that citizens

have more diverse activities on weekends than on weekdays. Then, the distribution of travel distance shown in Figure 14a,c demonstrates that human travel of different frequent interaction patterns still conforms to a power-law distribution. As Figure 14b,d shows, the cumulative frequency curves of travel distance show that the frequency of travel distance within 15 km of functional interaction patterns on weekdays (except for {R → T}, {A, R → T}) and weekends (except for {A, R → T}) is higher, indicating that citizens are more likely to take a taxi within 10 km for different travel intentions. As for {R → T}, {A, R → T} patterns on weekdays and {A, R → T} patterns on weekends, the travel distance reaches an inflection point at 25 km, showing that citizens have a high tolerance for longer trips when they travel to or from a transportation hub by taking a taxi.

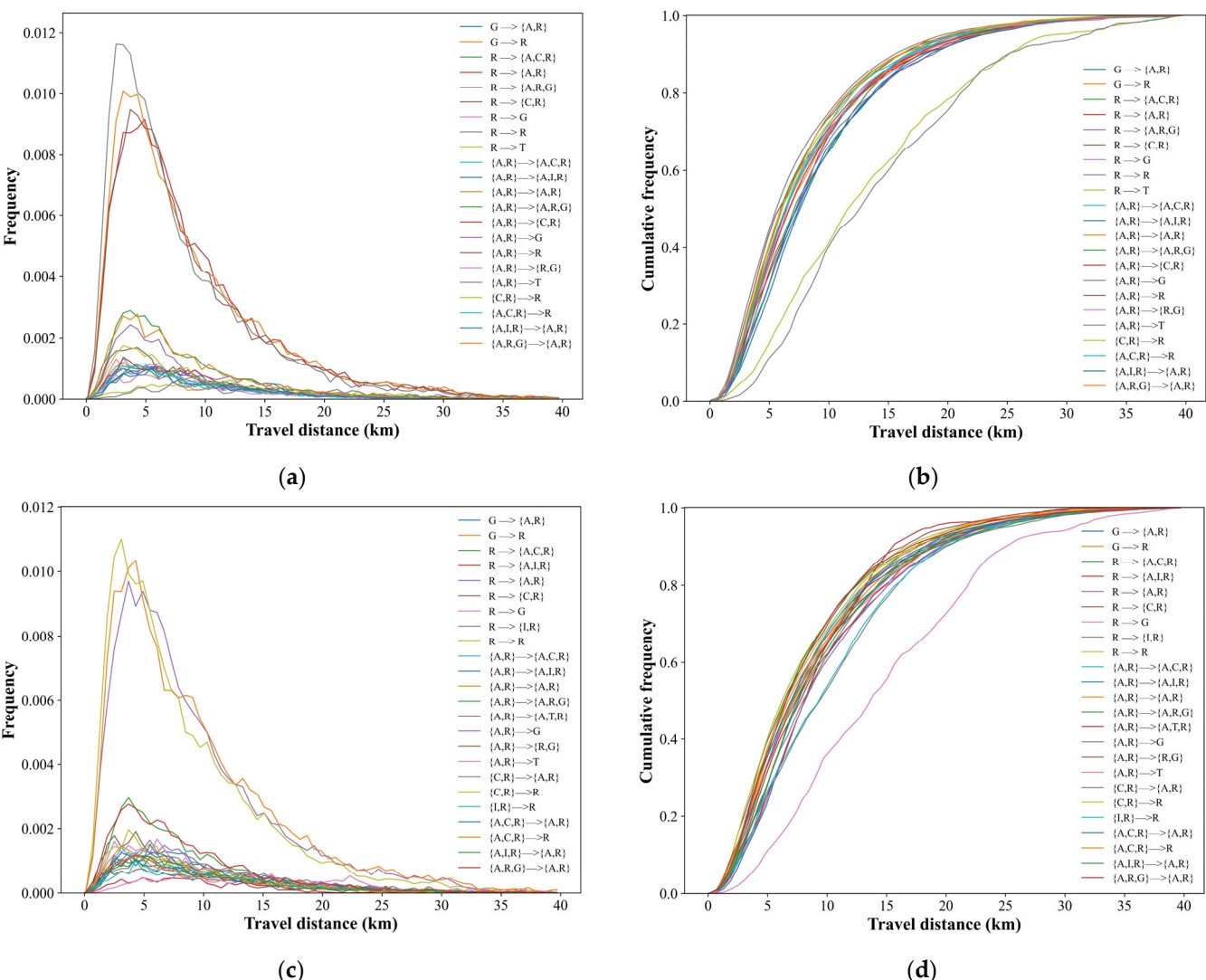

**Figure 14.** Travel distance characteristics of the frequent interaction patterns of UFZs on weekdays and weekends: (**a**) travel distance distribution of the mined 22 frequent interaction patterns on weekdays; (**b**) the cumulative frequency curve on weekdays; (**c**) travel distance distribution of the mined 24 frequent interaction patterns on weekends; (**d**) the cumulative frequency curve on weekends.

The travel time characteristics of frequent interaction patterns are shown in Figure 15. It has some similar characteristics to travel distance. First, the distribution of travel time for different frequent interaction patterns also conforms to the power-law distribution with different parameters. The travel time of almost all frequent function interaction patterns on the weekend is slighter longer than that on the weekday through the holistic analysis of travel time. The inflection point of travel time for all frequent interaction patterns on the

weekend is 35 min. The inflection point of travel time for {R → T} and {A, R → T} patterns on the weekday is about 40 min, which is about 10 min longer than the travel time of the other frequent interaction patterns on the weekday. The travel time of these two patterns is also 5 min longer than those two patterns on the weekend. The reason for this can be attributed to the phenomenon that traffic is more congested on the weekday than on the weekend, especially in the morning and evening peak hours on the weekday.

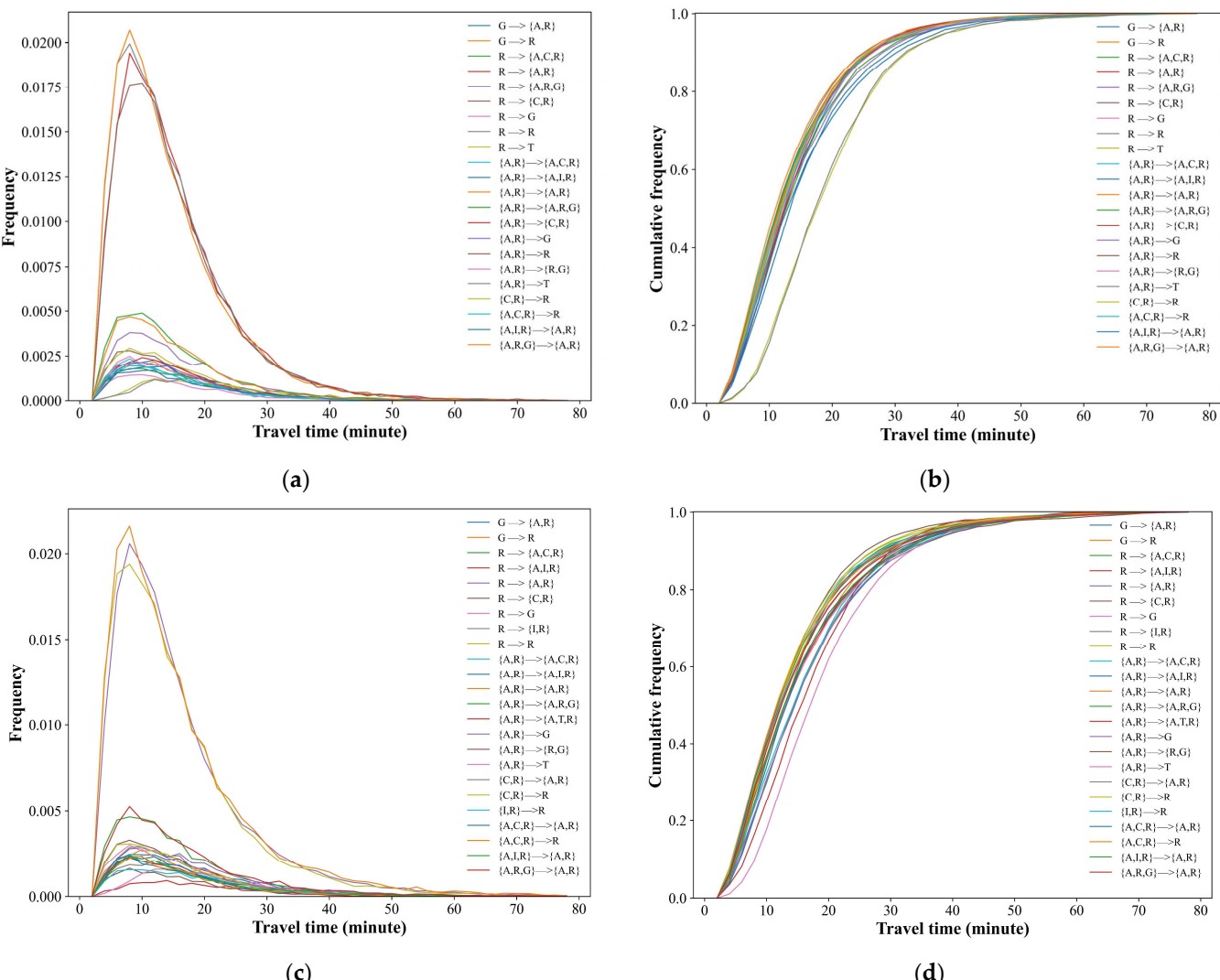

**Figure 15.** Travel time characteristics of the frequent interaction patterns of function zones on weekdays and weekends: (**a**) travel time distribution of the mined 22 frequent interaction patterns on weekdays; (**b**) the cumulative frequency curve on weekdays; (**c**) travel time distribution of the mined 24 frequent interaction patterns on weekends; (**d**) the cumulative frequency curve on weekends.

## 6. Conclusions

In this study, we give a case study of spatiotemporal interaction pattern analysis in Wuhan, China, based on urban functional zones. We first constructed the interaction matrix of functional zones and analyzed the interaction characteristics. Then, we detected the hotspots of functional zones, and the results show that the areas where most people gather are usually mixed functional zones, which may provide more services. We further extracted the frequent interaction patterns of functional zones in different time periods. Then we calculated the characteristics of travel time and travel distance of the frequent interaction patterns, and finally explored the underlying travel demands involved in crowd travel patterns.

Discovering the spatiotemporal interaction patterns of citizens' daily travel from the perspective of functional zones could provide support to the applications of urban planning and traffic control. Firstly, based on the characteristics of citizens' travel patterns discovered in this paper, more targeted and detailed traffic control policies can be made on weekdays or weekends and during different time periods in the day. For example, it could be possible to dynamically adjust the departure frequency of public transport according to citizens' traffic flows in different time periods. Secondly, to make the urban spatial structure better and more rational, it is recommended to appropriately exploit urban cold spots and plan cold-spot areas that may exist between frequently interactive functional zones according to the detected cold spots and hot spots and discovered frequently interactive functional areas in urban space. In addition, the research in this paper can also be generalized to any city in China and provide constructive suggestions for the city based on the corresponding results.

In our future work, the proposed method will be further investigated and tested with more extensive data and case studies. Data from multiple sources such as mobile phones, social media, IC cards, and surveying data of geographical environments bearing human activities can be integrated and used to achieve a more comprehensive understanding of human travel patterns.

**Author Contributions:** Conceptualization, Ju Peng, Huimin Liu, Jianbo Tang and Min Deng; Data curation, Ju Peng and Yiyuan Xu; Formal analysis, Ju Peng, Huimin Liu, Jianbo Tang, Xuexi Yang and Yiyuan Xu; Funding acquisition, Huimin Liu, Jianbo Tang; Investigation, Ju Peng, Huimin Liu, Cheng Peng and Yiyuan Xu; Methodology, Ju Peng, Huimin Liu, Jianbo Tang, Cheng Peng and Yiyuan Xu; Supervision, Huimin Liu, Jianbo Tang, Xuexi Yang and Min Deng; Writing—original draft, Ju Peng; Writing—review & editing, Ju Peng and Jianbo Tang. All authors have read and agreed to the published version of the manuscript.

**Funding:** This work was funded by the National Natural Science Foundation of China (grant numbers 42271462, 42171441, and 42271485); Hunan Provincial Natural Science Foundation of China (grant numbers 2021JJ40727, 2020JJ4749, and 2022JJ40585); the Scientific Research Project of Natural Resources Department of Hunan Province (grant numbers 2013-17, 2014-12, 2015-09, and 2017-15); the Fundamental Research Funds for the Central Universities of Central South University (grant number 2022zzts105); the Hunan Provincial Innovation Foundation for Postgraduate (grant number CX20220193).

**Data Availability Statement:** Not applicable.

**Acknowledgments:** This work was conducted in part using computing resources at the High-Performance Computing Platform of Central South University.

**Conflicts of Interest:** The authors declare no conflict of interest.

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
