# Peer review of "Exploring Crowd Travel Demands Based on the Characteristics of Spatiotemporal Interaction between Urban Functional Zones"

_ijgi, doi:10.3390/ijgi12060225_

Round 1

Reviewer 1 Report (Previous Reviewer 2)

The authors heavily revised the paper taking into consideration all the comments provided. While my concerns about the technical novelty of the work (clearly) remain, the overall quality of the paper improved significantly.

The new written parts of the paper should be revised, as they contain several typos and unclear expressions. Some examples are given below.

- Definition 1 (line 268-) should be rewritten. Simply state that itemsets can be formed by a single function tuple or by a pair of them.

- Line 273: "that with the same" is used several times, and seems incorrect. I suggest "that belong to the same".

- Formula (5) contains "P(A|B)", which is wrong. Please, simply remove that part.

- Lines 299- : did you really apply the Apriori algorithm? My understanding was that you directly computed all supports, confidences and lifts from the matrix.

- Page 9: the trajectories run from May 8 to 14, yet the weekday data is from May 6. How can that be?

-

- Line 193-194: "the larger of the ... the more related of the", please remove "of" in both positions.

- Line 196 is broken.

- Line 221: "gathering" -> "gather"

- Title of section 5.1: "maxtrix"

- Line 464: "the detected the" -> remove last "the"

- Line 506: "ap-pear"

Author Response

Reviewer 2 Report (Previous Reviewer 3)

The paper is in a better condition compared to the previous version. 

The minor problem with the paper is its mathematical presentation. Some formulas are categorized as common knowledge (primary probability equations) or represent heuristics. I suggest the authors further improve the mathematical formulas.

The observations represented by various plots are this paper's main point of strength.

Finally, the Authors would better check the spelling and grammatical issues.

Author Response

Reviewer 3 Report (New Reviewer)

Dear colleagues, thank you for your interesting approach in your contribution of exploring crowd travel demands based on the characteristics of spatio-temporal interaction between urban functional zones. 

By reading and analysing your paper I have observed three partly new perspectives: the analysis of crowd travel demands by spatiotemporal interaction patterns, the identification of contextual semantic features and the assessment of the GI* statistical spatial hotspot detection algorithm. All of these are valuable insights and approaches that need further investigation in other testing areas, with more extensive data and various use cases. 

Also some question came up when reading the paper:

In line 162 you name functional zones. How did you identify these functional zones? By definition? Then a reference will be needed in this section. Or by assessment/definition?

In line 178 you add semantic features of urban functionalities. How do you define these semantic features? How does the ontology for your use case look like?

In section 4 starting in line 338 you describe the data of your study. Here then realizes that the travelling data are restricted to the taxi mode. This is a very clear focus within public transport and should be widened, as mentioned in the conclusion. Also the use of openstreetmap data leads to the question if reliable authoritative data are missing?

I have enjoyed reading this paper and its approach, although having some flaws. Thank you.

Author Response

This manuscript is a resubmission of an earlier submission. The following is a list of the peer review reports and author responses from that submission.

Round 1

Reviewer 1 Report

Perhaps place a caveat in the discussion to be aware that urban planning by discrete functional zones is in many cases being replaced by more mixed use planning and transit oriented development.

Reviewer 2 Report

The paper is basically a follow-up of [39], where the Wuhan main area was partitioned into geographical units showing some characteristic function. In this work, a set of network properties of resulting graph and flows properties are analyzed with simple means, trying to identify hotspots, patterns and relations with other features.

The technical novelty looks rather limited -- and I believe there are also a few flaws, discussed below --,  thus the main interest of the paper appears to be in the use case. However, most of the findings did not go beyond what common sense or other, much simpler information sources could provide. Detailed comments follow.

- Abstract and later on: a "geographical detector" is mentioned a few times, before ever explaining what it is.
- Section 3, general: the approach assumes that the geographical units are meaningful functional areas. Is that the case with the Wuhan dataset? What happens if functions are mostly mixed? Does the approach have a clear meaning?
- Line 188: "to the maximum theoretical value" should actually read "... theoretical number of connections among the neighbors of the block".
- Line 204: "their nearest functional zones": considering that zones are a partitioning of the whole area, it should read "the functional zones the fall in".
- Line 211: "long-term" seems a misleading term to me. Maybe "long-range"?
- Line 220: the section mentions the interaction strength, which is not clearly defined. Actually, since the previous section was on connectivity measures, the text seems to suggest that the two concepts (strenght vs connectivity) are equivalent -- which is not the case.
- While I understand the idea behind Formula (4), it seems to grow with the area of the zone measured (more exactly, linearly with the perimeter), which seems strange for a compactness measure. Some comment would be welcome.
- Formula (7): I guess that S represents a standard deviation, in which case the last term (barred X) should be squared.
- The approach described in Section 3.3.2 is a simplified version of the classical Apriori algorithm. I suggest to simply adopt (there are dozens of extremely efficient implementations) and reference it, instead of introducing a new algorithm.
- Ibidem: Just using the support of the itemset as quality measure is very much limiting and biased, since high-frequency functions will always emerge over the others. I believe that is the reason why Figure 6 looks rather "flat" and slightly uninteresting. More refined measures should be adopted, e.g. the Lift or Interest ones.
- Section 4: the paper does not provide details about the dataset, just referencing [39]. To be self-contained, it should mention basic information, like the number of trips in the dataset, how POIs were used / selected, and in general how it was preprocessed.
- Ibidem: The dataset used is rather limited. Beside the fact that just a single area is considered (thus leaving space to questions about its replicability elsewhere), it only uses 2 days of trip data, whereas it is quite normal to have importat variations in mobility during the year. How much do those two days represent the local mobility? Without an answer to this, the results are kind of unreliable.
- Line 346: Section numbering seems to be wrong.
- Ibidem: this section should be about global analysis, yet it starts with a local one. Please, remove the incoherence.
- Figure 7 is not very informative. How is the block function represented? Are they weekday vs. weekend? Why is the spatial layout different? The centroid coordinates should be used to position nodes, in order to make the two plots comparable.
- Table 1: the differences between weekday and weekend are rather negligible. Maybe a different set of measures should be used.
- Line 407: what is the precise notion of "strength of spatial interaction" used here? Sum of inflow and outflow of the block? This part of the terminology remains vague throughout the paper.
- Table 2: what is the threshold that makes a q-statistic significant?
- Figure 9 and discussion in the text: the R^2 statistics seems very low to me, making the curve fitting not so significant. The conclusions drawn might be not well supported.
- The results described in section 5.3.1 seem to just confirm common sense. Are there other insights? The same applies to 5.3.2.
- Figures 12 and 13 are nice to see, but very little readable and informative -- quite common with chord diagram. Maybe a table reporting numerical values with colors might provide a better visions of what is happening.
- Figures 14-16: first, captions of the vertical axis is in most case wrong. Second, frequencies should probably be normalized (i.e. showing a relative one) in order to emphasize the shapes of distributions, rahter than their magnitude (which is more or less alrady known).

Minor issues (typos, etc.)
- Abstract, line 11: missing ")"
- Line 31: "needs" sounds a bit strange to me. Maybe "conditions"?
- Line 149: inter -> infer
- Line 169: "connection degree of the nodes" suggests the standard node degree, whereas the exaple suggests it is a matter of flow.
- Line 244: the reference (6) seems a typo.
- Section 3.3.1: the title would sound better to me without the "of"
- Line 300: covert -> convert
- Figure 11: the caption should mention "weekend"

Reviewer 3 Report

The article proposes a framework to discover the spatiotemporal interaction patterns in urban functional zones, from the perspective of spatial flow. The work considers the origin-destination (OD) matrix and analyzes the strength of the spatial interactions and travel patterns.

Authors need to put a good effort to increase the readability of the paper, especially in the abstract and introduction sections. Maybe a diagram or observation can improve the introduction section.

Example:

Further, functional zone pairs with frequent strong interactions in different periods are extracted to reveal local spatial interaction patterns using hotspot detection and frequent pattern mining method.

Firstly you need to alter the term “further” with “furthermore”. The sentence can be written shorter. Also do not write the sentences in an objective way. E.g. different periods are extracted, the authors would better say: We extract different periods or Our model leverages various periods.

The introduction section must highlight the challenges in a more readable way and then propose the solution.

One main missing area in this work is the fact that travel patterns change, affected by time. Therefore, it is required that authors include the related work in the introduction. Example 1: Mining subgraphs from propagation networks through temporal dynamic analysis. and Example 2: Leveraging multi-aspect time-related influence. 

Consider this important point the OD matrix can be different in various hours, days, or special events. You would better discuss this concept.

Also, we can handle OD matrices using matrix factorization, SVD, and deep learning methods.

The authors would better discuss the scientific contributions of the framework in the methodology. At times the model seems to be designed using heuristics, referring to Equations 1, 2, and 3. Hence, the authors would better utilize newer analytical approaches to better demonstrate the insights about influencing factors of the spatial interactions, Machine learning and in general AI approaches or genuine trajectory modeling techniques are standard directions for future work.

Referring to figure three the authors must elucidate how frequently the hot spots must be detected. This is related to the concept of triggering where the model needs to be refreshed, known as bilateral propagation graphs in the related work.

The experiment section needs to be designed with a better structure. The baselines and the benchmark are the two main parts to be included.

The time can be modeled using a discrete or continuous manner. It would help if you discuss both, e.g. Value-wise ConvNet for Transformer models is an elegant work in IEEE TKDE, explaining how we can continuously model the temporal information. The patterns may evolve and the formula may change.  The travel patterns of the citizens can change over time, this can be modeled using embedding approaches to be discussed in the related work: e.g. 1. Soulmate as a Multi-aspect temporal-textual embedding approach and 2. TEAGS: time-aware text embedding approach to generate subgraphs